# Study on the Method of Household Waste Collection: Case Study

**Mirela Panainte-Lehadus** [1] , **Mihai Vulpe** [1] , **Valentin Nedeff** [1,2] , **Emilian Mosnegutu** [1,*] , **Grzegorz Przydatek** [3] , **Claudia Tomozei** [1] and **Dana Chitimus** [1]

1   Faculty of Engineering, "Vasile Alecsandri" University of Bacău, 600115 Bacău, Romania; mirelap@ub.ro (M.P.-L.); vulpemh@yahoo.com (M.V.); vnedeff@ub.ro (V.N.); claudia.tomozei@ub.ro (C.T.); dana.chitimus@ub.ro (D.C.)
2   "Gheorghe Ionescu-Şişeşti" Academy of Agricultural and Forestry Sciences, 011464 București, Romania
3   Institute of Engineering, Nowy State University of Applied Sciences Sacz, Zamenhofa 1a, 33-300 Nowy Sacz, Poland; g.przydatek@gmail.com
*   Correspondence: emos@ub.ro

**Abstract:** This article presents research on how household waste is collected. An online survey, with 348 participants, from the Romanian region of Bacau, was conducted from October 2018 to May 2019. The online questionnaire included a set of over 40 questions, some with the aim of identifying the nature of the people participating, but most of the questions being designed to determine the collection methods for household waste. The major goal of the current study, as previously stated, was to determine the primary way of collecting household garbage from the public, while also learning various details about the participants, including their residence location, gender, age, and level of education. Referring to the means used for collecting household waste, the following items were noted: trash cans, cardboard boxes, dumpsters, and raffia bags. As a result of the study carried out, the following conclusions were drawn: it was noticed that 70 percent of those who participated in the survey came from urban areas; a larger percentage of female respondents took part in the survey (128 from 348); the majority of respondents were aged 18–29 (182); 178 respondents had a higher education level; collection of household waste in garbage bags represented 62.9 percent of the total collection methods. Following statistical processing of the data, and an overview of the main ways in which household waste was collected, a number of connections were found between the characteristics of the respondents and their household waste collection. What is noteworthy is that the characteristics of the respondents could be grouped into cumulative factors that played an important role in household waste collection: the first group formed by level of education and location of the respondents, and the second group formed by age and gender of the respondents.

**Keywords:** waste management; online questionnaire; household waste; statistical analysis



## 1. Introduction

With human progress, an increasing amount of waste is produced, resulting in waste disposal using up land, as well as contributing to global warming and having negative environmental impacts [1–3].

As a result of these factors, numerous concepts for reducing the quantity of waste created by both the public and enterprises have been developed.

Economic agents in Romania may restrict the quantity of trash created. At the European Union level, there are several legislative mechanisms that can be used to enforce existing laws. Even though the same measures apply to economic agents and to the population, the enforcement of the measures is difficult to achieve in the case of the general population.

The idea of sustainable development is more present in every aspect of human society. One of the components of this concept is the control of waste produced by both the populace and economic agents [4–15].

To create an overview of the concept of waste management as it is understood by the populace, a number of studies have been carried out by various governmental or commercial organizations [16–23]. In these studies, a precise focus is placed on aspects related to demographic parameters, the level of education of the population in the reference region, the type of housing, as well as the age of the population. The intention is to help the populace make the best waste management system choices. The choices about waste management methods are also known to be influenced by social behaviors, financial factors, cultural factors, etc. [24–37].

These studies were based on statistical analysis or real-world case analysis (for cities, institutions, areas with specific characteristics, etc.) [14,38–55].

In the future, we must ensure that people use existing waste management systems and also make use of specific systems to generate energy and material resources. To this end, people must be informed, and educational mechanisms must be established, that emphasize the importance of such ways of thinking [14,56–58].

This article describes online research on the temporary storage of household waste among the population of the Bacau region of Romania. The main purpose of this article was to identify how the population collects household waste. Whether there were statistical relationships between the different criteria used to define the group of respondents in this study and how they collect household waste was a further aim.

## 2. Research Methodology

Advances in software systems and current technological opportunities have made it possible to conduct targeted research on specific segments of the population at a faster rate and at a lower cost, while also providing many options for database design and interpretation.

The study presented in this paper was carried out using the Google platform, with 40-item questionnaires completed and distributed online.

This article offers the results of an analysis of part of the research. The Bacău region had a population of 742,053 in 2018. The data was provided from the responses of 348 participants who participated in the survey. Of the responses, 271 corresponded to a 90 percent confidence level [59], which, as a consequence, meant the study had a high degree of validity.

The questionnaire was distributed in an online environment and the results were obtained between 19 October 2018 and 1 May 2019.

The data flow diagram in Figure 1 shows the study's working process.

Based on the results of this study, it was decided to analyze the methods of waste collection at the generator's source, as well as the methods of waste collection by the general public, based on:

- Respondents' reference: urban area and rural area
- Gender: male and female
- Age: 15–17 years; 18–29 years; 30–39 years; 40–49 years; 50–59 years; 60–64 years; over 65 years
- Level of studies: secondary; university; postgraduate; other studies; professional

The collected material was subjected to descriptive and statistical analysis. Multi-way tables and homogeneity tests were used. The test probability value was $p < 0.05$. Frequency and multi-partition tables show the distribution of observations, due to several features at the same time. Pearson's chi-square test of independence and ML (Maximum Likelihood) for multi-way tables were used, these being used primarily in the study of the interdependence of nominal variables. In addition, hierarchical cluster analysis was applied using the agglomeration method, taking into account Ward's method and the percentage discrepancy matrix. In the article degrees of freedom (df) were tagged.

The data collected from the surveys was used to see if there were any relationships between the so-called input parameters and the home waste collection methods. Statistical analysis was performed for this purpose using two programs, namely StatSoft Poland and OriginPro.

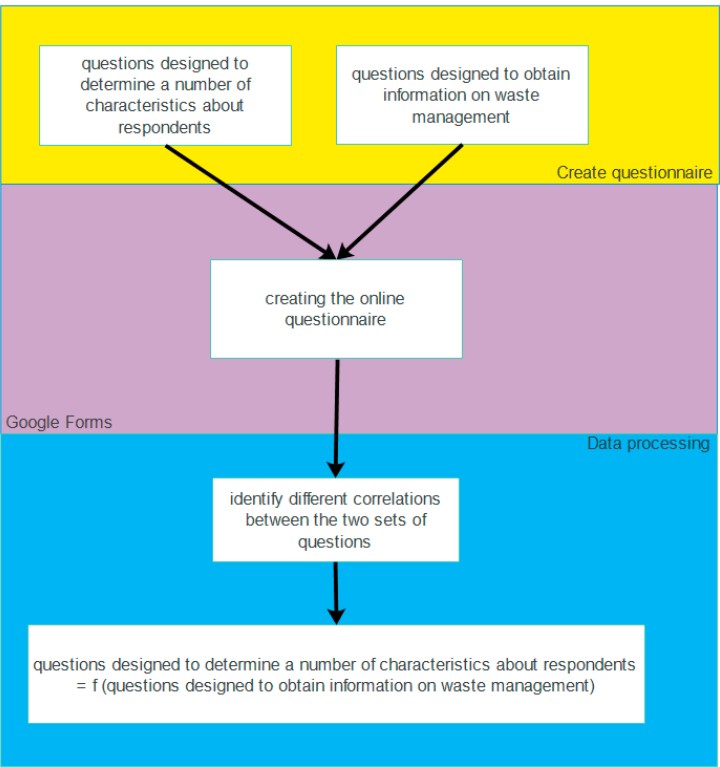

**Figure 1.** Working methodology.

## 3. Results

Following the analysis of the data obtained through the questionnaires, the following graphical representation was derived (Figure 2 and Table 1).

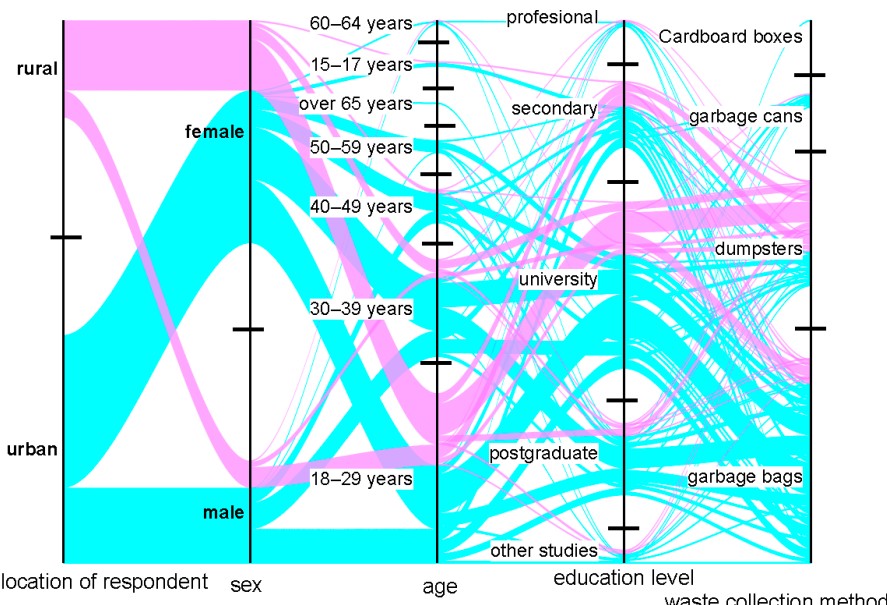

**Figure 2.** Distribution of the results obtained through the questionnaire.

**Table 1.** Table indicating the studied parameters' frequency analysis by classes, amount and waste collection method.

| Class | Number | Contribution [%] |
|---|---|---|
| Location of respondent (pcs.) | | |
| urban | 244 | 70.11 |
| rural | 104 | 29.89 |
| Sex (person) | | |
| female | 110 | 31.61 |
| male | 238 | 68.39 |
| Age (person) | | |
| 18–29 years | 182 | 52.30 |
| 30–39 years | 104 | 29.89 |
| 40–49 years | 36 | 10.34 |
| 50–59 years | 14 | 4.02 |
| 15–17 years | 6 | 1.72 |
| 60–64 years | 5 | 1.44 |
| above 65 years | 1 | 0.29 |
| Education level (person) | | |
| university | 178 | 51.15 |
| postgraduate | 77 | 22.13 |
| secondary | 71 | 20.40 |
| other studies | 15 | 4.31 |
| professional | 7 | 2.01 |
| Waste collection method (pcs.) | | |
| garbage bags | 219 | 62.93 |
| dumpsters | 113 | 32.47 |
| garbage cans | 16 | 4.31 |
| cardboard boxes | 1 | 0.28 |

The following conclusions may be drawn from the graphic representation in Figure 2:

(a)  70% of those who participated in the survey were from urban areas, with the rest from rural areas;

(b)  The number of female respondents who completed the questionnaire was determined to be 128 greater than the number of male respondents who did so;

(c)  The following results were reached after evaluating the age distribution of respondents who completed the questionnaire:

- The age group 18–29 years had the most respondents (182), with 57.7% from urban areas and 42.3% from rural areas;
- People aged 30 to 39 were the second largest category, in terms of the number of respondents, with 104, more precisely 84 from urban areas, and 20 from rural areas;
- The age groups with the lowest values corresponded to intervals of more than 65 years. Six respondents were found between the ages of 15 and 17, 36 respondents were found between the ages of 40 and 49, 14 respondents were found between the ages of 50 and 59, and for the age range of 60–64 years, just one responder was found.

(d)  An examination of the responses was also carried out, taking into consideration educational level of respondents. The following findings resulted from the investigation:

- There were 178 respondents with higher education, with 122 from the urban areas and 56 from the rural areas;
- There were 77 respondents with postgraduate degrees, 63 of them living in urban areas and 14 in rural areas;

- Those with a high school diploma had a score of 71, with 62% coming from urban areas and 38% living in rural areas;
- Other types of education applied to the remaining 3.73% of those who took part in the study (primary, secondary, vocational, or other).

(e) Several correlations between the characteristics of the studied group of respondents and the method of primary collection of household waste could be identified in a general analysis of the graphical representation in Figure 2. After the analysis, it was determined that:

- Many urban women between the ages of 18 and 29 and between 30 and 39 who have finished their university and postgraduate degrees collect household waste in household bags;
- Most urban men between the ages of 18 and 29 and between 30 and 39, with a high school or university degree, collect home waste in garbage bags;
- The majority of women in rural areas, between the ages of 18 and 29, with college degrees, collect household waste in landfills;
- In rural areas, most men between the ages of 18 and 29 with a high school education are responsible for collecting household waste in the trash.

In Table 1 it is noticeable that the largest share of 70.11% belonged to urban respondents. Men constituted the overwhelming share of 68.39% of the respondents. People aged 18–29 years constituted 52.30% of the respondents. It should be noted that respondents with higher education had a dominant share of 51.15%.

Next, a statistical evaluation of the data collecting method was carried out, based on the respondents' location (Table 2) and age (Table 3).

**Table 2.** Results of multi-partition table of method of waste collection taking into account the location of respondents.

| Location of Respondent | Waste Collection Method/Contribution [%] | | | Total |
|---|---|---|---|---|
| | Garbage Bags | Dumpsters | Garbage Cans | |
| Urban | *79.01%* | *15.23%* | *5.76%* | 70.03% |
| Rural | *25.96%* | *73.08%* | 0.96% | 29.97% |
| Total | 63.11% | 32.56% | 4.32% | 100.00% |

Italic value of statistics means that quantity of cells > 10.

**Table 3.** Multi-partition table of method of waste collection taking into account respondents' age.

| Age | Waste Collection Method/Contribution [%] | | | Total |
|---|---|---|---|---|
| | Garbage Bags | Dumpsters | Garbage Cans | |
| 18–29 years | 54.40% | 41.21% | 4.40% | 52.45% |
| 30–39 years | 73.08% | 23.08% | 3.85% | 29.97% |
| 40–49 years | 74.29% | 17.14% | 8.57% | 10.09% |
| 50–59 years | 71.43% | 28.57% | 0.00% | 4.03% |
| 15–17 years | 33.33% | 66.67% | 0.00% | 1.73% |
| 60–64 years | 100.00% | 0.00% | 0.00% | 1.44% |
| below 65 years | 100.00% | 0.00% | 0.00% | 0.29% |
| Total | 63.11% | 32.56% | 4.32% | 100.00% |

Based on this analysis, it was concluded that the collection of urban household waste in household bags accounted for the biggest percentage, at 79.01%. A significantly high share was collected in dumpsters in rural areas, with the lowest share (0.96%) pertaining to collection regarding garbage cans in the rural areas.

From the studied parameters, the location of the respondents, and the method of collecting household waste, had a Chi value of $Chi^2$ (Table 4).

**Table 4.** Chi-square statistics and its components taking into account the location of the respondents and waste collection methods.

| Statistic Location of Respondent × Waste Collection Method | | | |
|---|---|---|---|
| **Statistic** | **Chi2** | **df** | ***p*** |
| Chi2 Pearson | 111.2061 | df = 2 | *p* = 0.0000 |
| Chi2 ML | 109.9521 | df = 2 | *p* = 0.0000 |

The statistically significant values for the Pearson and ML square had the same *p* = 0.000, which confirmed the significant influence of the respondents' locations on waste collection methods.

According to the analysis of the results, respondents between the ages of 40 and 49 collected the most waste into bags at 74.29%. In this mode of collecting, those aged 30–39 had a significantly lower 1.01%, and also the lowest percentage regarding use of dumpsters for waste collection.

The data from this group was subjected to analysis determining the value of the chi-square parameter (Table 5).

**Table 5.** Chi-square statistics and its components taking into account age and waste collection method.

| Statistic Age × Waste Collection Method | | | |
|---|---|---|---|
| **Statistic** | **Chi$^2$** | **df** | ***p*** |
| Chi$^2$ Pearson | 23.22422 | df = 12 | *p* = 0.02588 |
| Chi$^2$ NW | 25.98123 | df = 12 | *p* = 0.01080 |

Statistically significant values occurred when the Pearson Chi-square value did not exceed *p* = 0.05, which confirmed the influence of age on the collection method.

Next, we aimed to identify cluster correlations between the study input parameters and the output parameters.

In the beginning, a cluster analysis was performed using Ward's method. The result of this analysis is presented in Figure 3.

Based on the analysis of the two indicated clusters using Ward's method and the percentage of non-compliance matrix, it should be noted that the first cluster was dominated by rural respondents, men aged 18–29, with higher education, collecting waste, mainly in dumpsters. On the other hand, in the second group, the majority of respondents from the urban area, who also were men in the above-mentioned age group with the same level of education, collected garbage in bags.

Following this analysis, the next conclusions could be drawn:

- The highest share of used bags for segregation (79.01%) was in the urban area which underlined the importance of larger settlements in the efficiency of waste management;
- The significant influence on garbage collection method was age, which was confirmed by the highest share of 74.29% of respondents being aged 40–49 aged collecting in garbage bags;
- Men from the city and rural areas with higher education in the 18–29 age group have a significant impact on waste collection in Bacau County.

Figure 4 shows a Hierarchical Cluster Analysis. Cluster analysis was used to explain the detection of the structure in the data, taking into account the hierarchical grouping. This analysis allows the determination of the so-called tree structure of the elements of the analyzed object set.

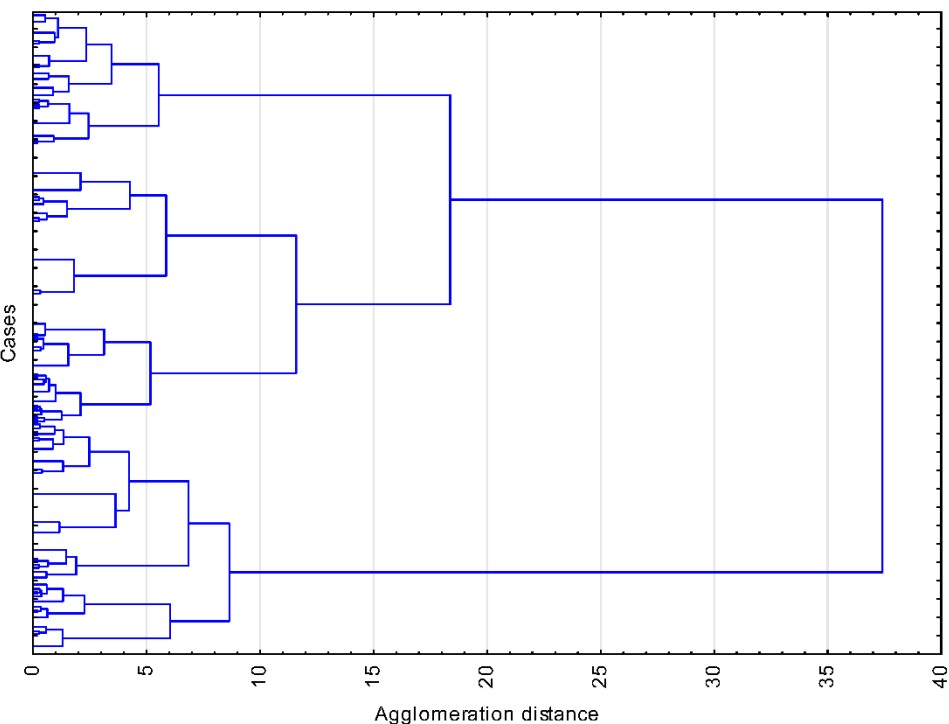

**Figure 3.** Agglomeration of Ward's method.

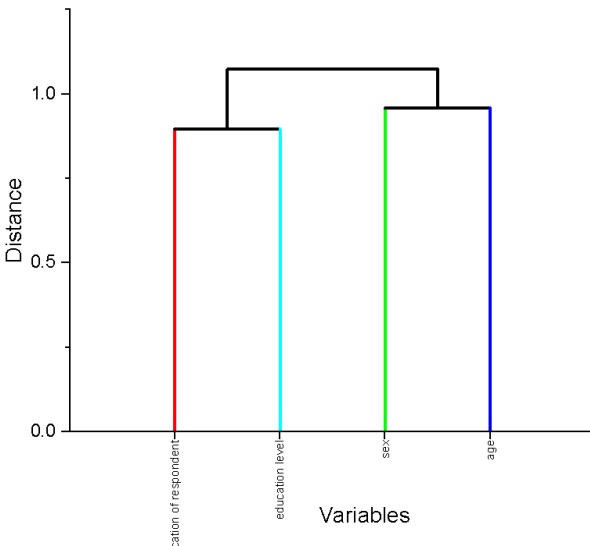

**Figure 4.** Analysis of hierarchical clusters.

Following this analysis, the following conclusions could be drawn:

- two clusters were identified, the first at a distance of 0.89, consisting of the level of education and location of the respondents, and the second cluster at a distance of 0.95, consisting of the respondent gender and age;
- the third cluster, which was secondary, connected the other two main clusters, the distance was 1.07;
- the last factor, with a distance of 1.08, was the location of the respondent.

## 4. Conclusions

This study was necessary because there are not many regional or national analyses of the solid household waste collection system which include the population that is served [47,53,54,59].

This study concentrated mostly on how the populace collects solid household waste inside the residential area. It did not consider the waste collection management system (from door to door or collection at a specified collection site) [3,11,12,19,23,28,32].

Following analysis of the results, it was found that:

○ The share of respondents corresponding to the urban area was 70.1%, and the respondents from the rural area accounted for 29.9%;

○ Regardless of the reference area of the respondents, the number of female respondents who participated in the study was 36.8% higher than the number of male respondents;

○ The best efficiency of waste management was felt in the urban area which was confirmed by the highest share of used bags for segregation (79.01%).

○ Taking into account the level of education of the respondents, 178 respondents had higher education, 77 respondents had postgraduate studies, and 71 respondents only had high school studies;

○ In terms of age, in this online survey, it was found that the largest share was held by respondents aged between 18 and 29, namely 182 respondents, followed by respondents aged between 30 and 39 years, numbering 104 respondents;

○ As a method of collection of household waste, it was found that the collection of waste in garbage bags accounted for 62.9%, followed by collection in dumpsters at 32.4%, and collection into garbage cans at 4.3%;

○ Statistical analysis was done to find the relationships between the study's input and output parameters. For this reason, a cluster statistical analysis was used. Following these analyses, a series of correlations were identified.

○ The identification of the method of primary collection of waste generated by the population aimed to achieve an overview of the understanding of the concept of collection by the population;

○ Waste management was significant for young women who come from a city with higher education;

○ In conclusion from the questionnaire, it can be said that the level of education and gender are the main factors that directly influence the collection of municipal waste, followed by the age of the respondent and the place where the respondent lives.

**Author Contributions:** Conceptualization, M.V.; methodology, V.N.; software, E.M. and G.P.; validate, V.N.; formal analysis, C.T. and D.C.; investigation, M.P.-L.; drafting—drafting the original project, M.V.; writing—review and editing, E.M. and G.P.; supervision, V.N.; project management, V.N. All authors have read and agreed to the published version of the manuscript.

**Funding:** This research received no external funding.

**Institutional Review Board Statement:** Not applicable.

**Informed Consent Statement:** Not applicable.

**Data Availability Statement:** Not applicable.

**Conflicts of Interest:** The authors do not declare any conflict of interest.

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
