# Peer review of "Study on the Method of Household Waste Collection: Case Study"

_applsci, doi:10.3390/app12157490_

Round 1
Reviewer 1 Report
The paper proposes a statistical study based on 619 people completing online questionnaires in order to correlate waste collection in households with age, gender, household reference area and level of education.
Unfortunately, the manuscript has mistakes and lacks explanations or delivers information hard to follow. Also, one of the concerns is related to the fact that the study was performed online, which implies that only internet owners were able to participate. In this case, how relevant the results are?
The study includes figures that repeat the same information given by the previous ones (Fig.7 conveys the same information as Fig 6, and both figures have the same data as in Figs.3-5). Fig.8 is also a 3D version of Figs. 4-5. Additionally, figures that do not have any description relevant to the study (Fig.9) are included, so their presence is not justified. Some of the references are not relevant to the study.
Thus, the manuscript should be thoroughly reviewed and go through extended modifications/corrections, possibly with new data added for a more substantial scientific content, before a new submission.

Author Response
Due to comments received from all reviewers, the article has been completely rewritten.
We apologize for not respecting the time to correct the article according to your comments.
Reviewer 2 Report
This paper presents the results obtained from the administration of a questionnaire on the primary collection method of waste. The work should be extensively reviewed from both a methodological and organizational point of view. Some suggestions:
- The abstract should be rewritten to report a description of the context, the objective of the work, a brief mention to the methodology adopted and the main results obtained.
- The introduction does not give any scientific relevance to the work. It should be rewritten, inserting data and references that allow the reader to understand the relevance of the topic. It should include the objective of the work, a mention of the methodology adopted and, in conclusion, a paragraph describing the structure of the work.
- The structure of the work should be rearranged in a more consistent way.
- The methodology should be set out clearly. At present, it is not understandable what has been done in the work. Figure 2 is unnecessary.
- The results should be reorganized by discussing them in a meaningful way.
- The methodology used to describe the correlation of the observed variables should be better developed, adding further evaluation.
- The conclusions should be rewritten in order to show meaningful considerations. Limitations of the present work and recommendations for future studies should be included in this section.
Author Response

(The authors gave the same response as above.)

Reviewer 3 Report
I have the following comments:
- The abstract is not informing about the certain results.
- Line 29 – 32: these sentences are without references.
- Line 36: the sentence is not written correctly; it should be revised.
- Line 43: what is the purpose of this sentence.
- Figure 1 is not necessary.
- The whole introduction part is written confusing and it should be revised.
- 619 respondents are not a big amount for the research.
- Socio demographic information about respondents should be given in the Table.
- Why authors did not use chi square test?
- Statistical analysis is not clearly presented and the discussion part is very small.
Author Response

(The authors gave the same response as above.)

Round 2
Reviewer 1 Report
The manuscript has been improved and the data analysis methodology has been changed. However, there are some issues, listed below that need further revision.
The review comments are enumerated below:
1. Figure 3 is related to Ward method. Please add some specifications about the x- and y- axes and some explanations that connect the graph to the conclusions (yellow field) that precedes it. What does the red line indicate? What was the maximum R-square?
2. In Table 5, please specify in the text what df and ML stands for.
3. Some references are not relevant for the study (for example 25).
4. Some English corrections are still necessary to be made.
Author Response
Thank you for your comments, which have the role of increasing the value of our article. The answers to your comments are marked in yellow.

Reviewer 2 Report
Thanks to the authors for following the suggestions provided. The work is now clearer and more complete. In the introduction, however, not enough scientific relevance is given to the topic, as well as the abstract could be improved and made clearer.
Author Response

(The authors gave the same response as above.)

Reviewer 3 Report
I have the following comments:
1. What is the statistical method applied on the study?
2. Table 1: the table is technically not well made. In all other tables authors should pay the attention on the text justification in tables.
3. Authors have all references in the Introduction part, Authors do not have references in the discussion part!!!!
4. Authors should also mention the edible oil as household waste since it is highly toxic for the environment. The following reference should be used: Antonić, B., Dordević, D., Jančíková, S., Tremlova, B., & Kushkevych, I. (2020). Physicochemical characterization of home-made soap from waste-used frying oils. Processes, 8(10), 1219.
Author Response

(The authors gave the same response as above.)
